# WASSERSTEIN PROXIMAL OF GANS

## ABSTRACT

We introduce a new method for training GANs by applying the Wasserstein-2 metric proximal on the generators. This approach is based on the gradient operator induced by optimal transport theory, which connects the geometry of the sample space and the parameter space in implicit deep generative models. From this theory, we obtain an easy-to-implement regularizer for the parameter updates. Our experiments demonstrate that this method improves the speed and stability in training GANs in terms of wallclock time and Fréchet Inception Distance (FID) learning curves.

## 1 INTRODUCTION

Generative Adversarial Networks (GANs) (Goodfellow et al., 2014) are a powerful approach to learning generative models. Here, a discriminator tries to tell apart the data generated from a real source and the data generated by a generator, whereas the generator tries to fool the discriminator. This adversarial game is formulated as an optimization problem over an implicit generative model for the generator. An implicit generative model is a parametrized family of functions mapping a noise source to sample space. In trying to fool the discriminator, the generator should try to recreate the density distribution from the real source.

The problem of matching a target density can be formulated as the minimization of a discrepancy measure. The Kullback–Leibler (KL) divergence is known to be difficult when the distributions have a low dimensional support set, as is commonly the case in applications with structured data and high dimensional sample spaces. An alternative approach to define a discrepancy measure between densities is optimal transport, a.k.a. Wasserstein distance, or Earth Mover's distance. This has been used recently to define the loss function for learning generative models (Montavon et al., 2016; Frogner et al., 2015). In particular, the Wasserstein GAN (Arjovsky et al., 2017) has attracted much interest in recent years.

Besides defining the loss function, optimal transport can also be used to introduce structures serving the optimization itself, in terms of the gradient operator. In full probability space, this is known as the Wasserstein steepest descent flow (Jordan et al., 1998; Otto, 2001). In this paper we derive the Wasserstein steepest descent flow for deep generative models in GANs. We use the Wasserstein-2 metric function, which allows us to obtain a Riemannian structure and a corresponding natural (i.e., Riemannian) gradient. A well known example of a natural gradient is the Fisher-Rao natural gradient, which is induced by the KL divergence. In learning problems, one often finds that the natural gradients can offer advantages compared to the Euclidean gradient (Amari, 1998; 2016). In GANs, because of the low dimensional support sets and the associated difficulties with the KL divergence, the Fisher-Rao natural gradient is problematic. Therefore, we propose to use the gradient operator induced by the Wasserstein-2 metric (Li & Montúfar, 2018a;b).

We compute the proximal operator for the generators of GANs, where the regularization is the squared constrained Wasserstein-2 distance. In practice, the constrained distance can be approximated by a simple neural network. In implicit generative models, the constrained Wasserstein-2 metric exhibits a simple structure. We generalize the metric and introduce the relaxed proximal operator for generators, which allows us to further simplify the computation. The resulting relaxed proximal operator involves only the difference of outputs, so that the proximal computation has very simple parameter updates. The method can be easily implemented and used as a drop-in regularizer for the generator updates.

This paper is organized as follows. In Section 2, we briefly introduce the Wasserstein natural gradient. A Wasserstein proximal method is introduced in Algorithm 1. In Section 3, we demonstrate the effectiveness of the proposed methods in experiments with various types of GANs. Section 4 reviews related work.

## 2   WASSERSTEIN PROXIMAL

In this section, we briefly present optimal transport and its proximal operator on a parameter space. We then apply them to the optimization problems of GANs.

### 2.1   WASSERSTEIN NATURAL GRADIENT

Optimal transportation defines a class of distance functions between probability densities. Given a pair $\rho_0, \rho_1 \in \mathcal{P}_p(\mathbb{R}^n)$ of probability densities with finite $p$-th moment,

$$W_p(\rho_0, \rho_1)^p = \inf \int_{\mathbb{R}^n \times \mathbb{R}^n} \|x - y\|^p \pi(x, y) dx dy, \tag{1}$$

where the infimum is over all joint probability densities $\pi(x, y)$ with marginals $\rho_0(x), \rho_1(y)$. In the literature (see Villani, 2009), $W_p$ is referred to as the Wasserstein-$p$ distance. In this paper, we focus on the case $p = 2$, and further denote $W_2$ by $W$.

Following Benamou & Brenier (2000), the Wasserstein-2 distance has a dynamical formulation as a trajectory transporting the initial density $\rho_0$ to the final density $\rho_1$ along a trajectory of minimal kinetic energy. The classic theory does not consider the setting where the density path is constrained to lie within a parametrized model. In the following we extend the classic theory to cover parameterized density models. Consider a parameterized probability $\rho(\theta, x)$, with parameter space $\Theta \subset \mathbb{R}^d$. Suppose that $\rho(\theta, x)$ is locally injective as a mapping from $\Theta$ to $\mathcal{P}_2(\mathbb{R}^n)$. Then the Wasserstein-2 metric function constrained to the parameter space is given as follows (see Li & Montúfar, 2018a).

**Theorem 1 (Constrained Wasserstein-2 metric)** *The constrained Wasserstein-2 metric function $d_W \colon \Theta \times \Theta \to \mathbb{R}_+$ has the following formulation:*

$$d_W(\theta_0, \theta_1)^2 = \inf \Big\{ \int_0^1 \int_{\mathbb{R}^n} \|\nabla\Phi(t, x)\|^2 \rho(\theta(t), x) dx dt \colon$$

$$\partial_t \rho(\theta(t), x) + \nabla \cdot (\rho(\theta(t), x) \nabla\Phi(t, x)) = 0,\ \theta(0) = \theta_0,\ \theta(1) = \theta_1 \Big\},$$

*where the infimum is among all feasible Borel potential functions $\Phi \colon [0, 1] \times \mathbb{R}^n \to \mathbb{R}$ and continuous parameter paths $\theta \colon [0, 1] \to \mathbb{R}^d$. Here $\nabla \cdot$ and $\nabla$ are the divergence and gradient operators over $\mathbb{R}^n$.*

We note that the constrained metric on parameter space can be different from the Wasserstein-2 distance on the full density set. The metric $d_W$ can be used to define a steepest descent optimization scheme. This can be formulated in two general ways.

One way is in terms of the corresponding Riemannian structure, i.e., an inner product between tangent vectors. A well known example is the Fisher natural gradient (Amari, 1998; 2016). The constrained Wasserstein-2 metric allows us to obtain a Riemannian metric structure, from which we obtain the following constrained Wasserstein-2 gradient. We also call it *Wasserstein natural gradient*.

**Theorem 2 (Wasserstein natural gradient)** *Given a loss function $F \colon \Theta \to \mathbb{R}$, the Wasserstein gradient operator is given by*

$$\nabla_\theta^W F(\theta) = G(\theta)^{-1} \nabla_\theta F(\theta),$$

*where $G(\theta) = (G(\theta)_{ij})_{1 \le i,j \le d} \in \mathbb{R}^{d \times d}$ is given by*

$$G(\theta)_{ij} = \int_{\mathbb{R}^n} \nabla\Phi_i(x) \nabla\Phi_j(x) \rho(\theta, x) dx.$$

*Here for each $i \in \{1, \cdots, d\}$, $\Phi_i \colon \mathbb{R}^n \to \mathbb{R}$ is a solution (up to additive constants) of $\frac{\partial}{\partial \theta_i} \rho(\theta, x) + \nabla \cdot (\rho(\theta, x) \nabla\Phi_i(x)) = 0$.*

Here $\nabla_\theta^W$ represents the natural gradient operator with respect to the constrained Wasserstein metric, $\nabla_\theta$ represents the ordinary Euclidean gradient operator, and $G$ is the matrix representing the Wasserstein Riemannian metric. The steepest descent flow is given by

$$\frac{d}{dt}\theta(t) = -G(\theta(t))^{-1}\nabla_\theta F(\theta(t)). \tag{2}$$

The corresponding gradient descent iteration (forward Euler method) satisfies

$$\theta^{k+1} = \theta^k - hG(\theta^k)^{-1}\nabla_\theta F(\theta^k),$$

where $h > 0$ is the step size. Often in practice, the computation of matrix $G(\theta)^{-1}$ is difficult.

The second way of obtaining a numerical scheme for equation 2 is in terms of the proximal operator. This is the backward Euler method, also named Jordan-Kinderlehrer-Otto (JKO) scheme (Jordan et al., 1998), which is given by

$$\theta^{k+1} = \arg\min_{\theta\in\Theta} F(\theta) + \frac{1}{2h}d_W(\theta, \theta^k)^2. \tag{3}$$

Here, at each step, the distance of the parameter update acts as a regularization to the original loss function.

Computing $d_W$ is also often challenging. However, we can approximate the $d_W$ distance locally by a second order Taylor expansion. This approximation is particularly tractable within the parameterized setting that we discussed above.

This allows us to derive other first order schemes, such as the Semi-Backward Euler method:

**Proposition 3 (Semi-Backward Euler method)** *The Semi-Backward Euler method for the gradient flow of loss function $F\colon \Theta \to \mathbb{R}$ is given by*

$$\theta^{k+1} = \arg\min_{\theta\in\Theta} F(\theta) + \frac{1}{h}\sup_\Phi \int_{\mathbb{R}^n} \Phi(x)(\rho(\theta, x) - \rho(\theta^k, x)) - \frac{1}{2}(\nabla\Phi(x))^2\rho(\theta^k, x)dx,$$

*where the supremum is taken over $\Phi\colon \mathbb{R}^n \to \mathbb{R}$ with sufficient regularity for the integral to be well defined.*

The Semi-Backward Euler method is often easier to approximate than the forward Euler method, because it does not require computing and inverting $G(\theta)$, and it is often simpler than the backward Euler method (JKO), because the constrained optimization over $\Phi$ is more tractable than the time-dependent constraint involved in computing $d_W$.

We implement the Semi-Backward Euler method in implicit generative model as follows. For each parameter $\theta \in \mathbb{R}^d$, let the generator be given by $g_\theta\colon \mathbb{R}^m \to \mathbb{R}^n; z \mapsto x = g(\theta, z)$. This takes an input noise prior $Z \sim p(z) \in \mathcal{P}_2(\mathbb{R}^m)$ to an output sample with density given by $X = g(\theta, Z) \sim \rho(\theta, x)$. Here $\mathbb{R}^d$ is the parameter space, $\mathbb{R}^m$ is the latent space, and $\mathbb{R}^n$ is the sample space.

In this case, the update in Proposition 3 forms

$$\theta^{k+1} = \arg\min_{\theta\in\Theta}\sup_\Phi F(\theta) + \frac{1}{h}\mathbb{E}_{Z\sim p(z)}[\Phi(g(\theta, Z)) - \Phi(g(\theta^k, Z)) - \frac{1}{2}\|\nabla_x\Phi(g(\theta^k, Z))\|^2].$$

In practice, we apply a neural network to approximate variable $\Phi$. See details in Appendix G.

## 2.2 REGULARIZATION ON GENERATORS

In fact, the constrained Wasserstein-2 metric in implicit generative models allows for yet a simpler formulation. This reformulation allows us to define the relaxed Wasserstein metric, and further introduces a simple algorithm for proximal operator on generators.

**Proposition 4 (Constrained Wasserstein-2 metric in implicit generative models)**

$$d_W(\theta_0, \theta_1)^2 = \inf \Big\{ \int_0^1 \mathbb{E}_{Z\sim p(z)}\|\frac{d}{dt}g(\theta(t), Z)\|^2 dt :$$

$$\frac{d}{dt}g(\theta(t), Z) - \nabla_x\Phi(t, g(\theta(t), Z)) = 0,\ \theta(0) = \theta_0,\ \theta(1) = \theta_1 \Big\},$$

*where the infimum is among all feasible Borel potential functions $\Phi\colon [0, 1] \times \mathbb{R}^n \to \mathbb{R}$ and continuous parameter paths $\theta\colon [0, 1] \to \mathbb{R}^d$.*

Here the constrained Wasserstein metric requires that the derivative of the generator $g$ w.r.t. $\theta \in \mathbb{R}^d$ be a gradient vector field of $\Phi$ w.r.t $x \in \mathbb{R}^n$. In other words, if we denote $x(t) = g(\theta(t), z)$, then

$$\frac{d}{dt}x(t) = \nabla_x \Phi(t, x(t)). \qquad \text{(Gradient constraint)}$$

The gradient constraint is satisfied if the sample space is 1 dimensional, i.e., $n = 1$. In general, this is not true. Here $\Phi(t, x)$ is the other function depending on the parameter space $\Theta$. Finding $\Phi$ involves computational difficulties. Fitting the gradient constraint is an open problem for the computations of Wasserstein proximal operator.

For simple computations, we withdraw the gradient constraint and consider a relaxed Wassersetin metric on parameter space:

$$d(\theta_0, \theta_1)^2 = \inf_{\theta(t)} \left\{ \int_0^1 \mathbb{E}_{Z \sim p(z)} \|\frac{d}{dt} g(\theta(t), Z)\|^2 dt : \theta(0) = \theta_0, \ \theta(1) = \theta_1 \right\}.$$

We approximate the relaxed Wasserstein proximal operator based on the new metric $d$ to obtain

$$\theta^{k+1} = \arg\min_{\theta \in \Theta} F(\theta) + \frac{1}{2h} \mathbb{E}_{Z \sim p(z)} \|g(\theta, Z) - g(\theta^k, Z)\|^2, \qquad (4)$$

where the infimum is among all feasible continuous parameter path $\theta : [0, 1] \to \mathbb{R}^d$.

In fact, when the sample space is high dimensional, i.e., $n > 1$, the above update is not exactly the Wasserstein proximal. Instead, it simply **regularizes the generator** by the expectation of squared difference in sample space.

---

**Algorithm 1** Relaxed Wasserstein Proximal, where $F_\omega$ is a parameterized function to minimize

**Require:** $F_\omega$, a parameterized function to minimize (e.g. Wasserstein-1 with a parameterized discriminator). $g_\theta$ the generator.
**Require:** $h$ proximal step-size, $B$ batch size.
**Require:** Optimizer$_{F_\omega}$ and Optimizer$_{g_\theta}$.
**Require:** max iterations, and generator iterations
    **for** $k = 0$ **to** max iterations **do**
        Sample real data $\{x_i\}_{i=1}^B$ and latent data $\{z_i\}_{i=1}^B$
        $\omega^k \leftarrow \text{Optimizer}_{F_\omega} \left( \frac{1}{B} \sum_{i=1}^B F_\omega(g_\theta(z_i)) \right)$
        **for** $\ell = 0$ **to** generator iterations **do**
            Sample latent data $\{z_i\}_{i=1}^B$
            $\theta^k \leftarrow \text{Optimizer}_{g_\theta} \left( \frac{1}{B} \sum_{i=1}^B F_\omega(g_\theta(z_i)) + \frac{1}{h} \|g_\theta(z_i) - g_{\theta^{k-1}}(z_i)\|^2 \right)$
        **end for**
    **end for**

---

### 2.3 ILLUSTRATION OF WASSERSTEIN PROXIMAL

We present a toy example to illustrate the effectiveness of Wasserstein proximal operator in GANs.

Consider a family of distribution with two weighted delta measures. Let $\Theta = \{\theta = (a, b) : a < 0, b > 0\}$, and define

$$\rho(\theta, x) = \alpha \delta_a(x) + (1 - \alpha)\delta_b(x),$$

where $\alpha \in [0, 1]$ is a given ratio and $\delta_a(x)$ is the delta function supported at point $a$. See Figure 2.3.

In this model, for a loss function $F : \Theta \to \mathbb{R}$, the proximal regularization is given as follows:

$$\theta^{k+1} = \arg\min_{\theta \in \Theta} F(\theta) + \frac{1}{2h} d(\theta, \theta^k)^2,$$

where $\theta = (a, b)$ and $\theta^k = (a^k, b^k)$. We check the following commonly used statistical distance (divergence) functions $d$ between parameters $\theta$ and $\theta^k$.

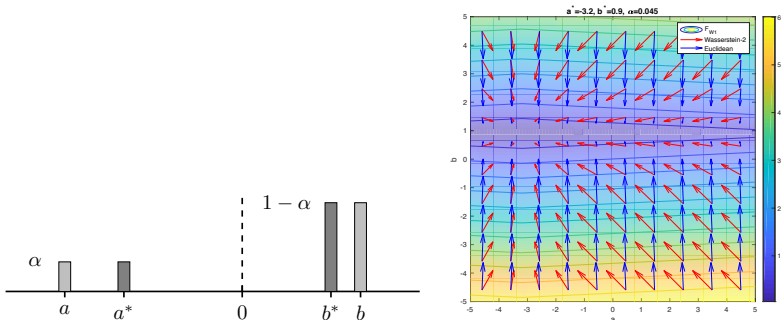

Figure 1: Illustration of the example from Section 2.3. The Wasserstein proximal penalizes parameter steps in proportion to the mass being transported, which results in updates pointing towards the minimum of the loss function. The Euclidean proximal penalizes all parameters equally, which results in updates naively orthogonal to the level sets of the loss function.

1. Wasserstein-2 distance:
$$d_W(\theta, \theta^k)^2 = \alpha(a - a^k)^2 + (1 - \alpha)(b - b^k)^2;$$

2. Euclidean distance:
$$d_E(\theta, \theta^k)^2 = (a - a^k)^2 + (b - b^k)^2;$$

3. Kullback–Leibler divergence:
$$d_{KL}(\rho_\theta \| \rho_{\theta^k}) = \int_{\mathbb{R}^n} \rho(\theta, x) \log \frac{\rho(\theta, x)}{\rho(\theta^k, x)} dx = \infty;$$

4. $L^2$-distance:
$$d_{L^2}(\rho_\theta, \rho_{\theta^k})^2 = \int_{\mathbb{R}^n} |\rho(\theta, x) - \rho(\theta^k, x)|^2 dx = \infty.$$

Here the KL divergence and $L^2$-distance cannot measure the difference of probability models. The Wasserstein-2 and Euclidean distances still work in these cases. In addition, the Euclidean distance $d_E$ does not depend on the structure of model $\rho(\theta, x)$, while the constrained Wasserstein-2 metric $d_W$ does.

**Proposition 5** *Given $\theta^* = (a^*, b^*) \in \Theta$, consider the Wasserstein-1 metric as the loss function, i.e.,*
$$F_{W_1}(\theta) = W_1(\rho_\theta, \rho_{\theta^*}) = \alpha|a - a^*| + (1 - \alpha)|b - b^*|.$$

*Denote $\theta_W^{k+1} = \arg\min_\theta F_{W_1}(\theta) + \frac{1}{2h} d_W(\theta, \theta^k)^2$, and $\theta_E^{k+1} = \arg\min_\theta F_{W_1}(\theta) + \frac{1}{2h} d_E(\theta, \theta^k)^2$. For each stepsize $h > 0$, then*
$$F_{W_1}(\theta_E^{k+1}) \geq F_{W_1}(\theta_W^{k+1}).$$

On each step of the update, the solution obtained by Wasserstein proximal decreases the objective function further than the one by Euclidean proximal. Here the proof is based on a simple fact of the shrinkage operator, see details in Appendix B.

This example introduces a case that Wasserstein-2 proximal works better than Euclidean proximal for the Wasserstein-1 loss function.

## 3 EXPERIMENTS ON GANS

Here we present numerical experiments using the Relaxed Wasserstein Proximal (RWP) algorithm and the Semi-Backward Euler (SBE) method in order to perform Wasserstein gradient-descent on various GANs. We find that the Relaxed Wasserstein Proximal provides both better speed (measured by wallclock) and stability in training GANs.

## 3.1 Results of Relaxed Wasserstein Proximal

The Relaxed Wasserstein Proximal (RWP) algorithm is intended to be an easy-to-implement, drop-in replacement to improve speed and convergence of GAN training. It does this by applying regularization on the generator during training. This is novel as most GAN training focuses on regularizing the discriminator, e.g. with a gradient penalty (Gulrajani et al., 2017b; Petzka et al., 2017; Kodali et al., 2018; Adler & Lunz, 2018; Miyato et al., 2018), and there has been limited exploration in regularizing the generator (Chen et al., 2016). Specifically, we modify the update rule for the generator by:

- Update for $\ell$ number of iterations before updating the discriminator:

$$\theta \leftarrow \text{Optimizer}_\theta \left( \text{Original loss} + \frac{1}{2h} \| g_\theta - g_{\theta^{k-1}} \|^2 \right)$$

So two hyperparameters are introduced: the proximal step-size $h$, and the number of iterations $\ell$. In some GANs, one may update the discriminator a number of times and then update the generator a number of times, and then repeat; we will call one loop of this update an *outer-iteration*.

A more detailed description of the algorithm is given in Appendix D.

We test the Relaxed Wassersteing Proximal regularization on three GAN types:

- Standard GANs (Goodfellow et al., 2014),
- WGAN-GP (Gulrajani et al., 2017a), and
- DRAGAN (Kodali et al., 2018).

We use the CIFAR-10 dataset (Krizhevsky, 2009), and the aligned and cropped CelebA dataset (Liu et al., 2015). And we utilize the DCGAN (Radford et al., 2015) architecture for the discriminator and generator. To measure the quality of generated samples, we employ the Fréchet Inception Distance (FID) (Heusel et al., 2017) both to measure performance and to measure convergence of GAN training (lower FID is better); we used 10,000 generated images to measure the FID. For CIFAR-10, we measure the FID every 1000 outer-iterations, and for CelebA we measure the FID every 10,000 outer-iterations.

Our particular hyperparameter choices for training are given in Appendix C. Note that since we intend RWP to be a drop-in regularization, the non-RWP hyperparameters (i.e. not $h$ nor $\ell$) are chosen to work well before applying RWP.

To summarize our results, the Relaxed Wasserstein Proximal regularization improves both the speed (wallclock) and stability of convergence. It is a tricky to compare the result of using RWP, as it performs multiple generator iterations. We thus align the comparison according to wallclock time (this procedure was also used by Heusel et al., 2017). In Figure 2 we see that our regularization improves convergence speed (measured in wallclock time), and also obtains a lower FID for all GAN types. In particular, in DRAGAN we see a 20% improvement in sample quality according to the FID. The same results are also found for the CelebA dataset, shown in Figure 3. We note that multiple generator iterations will sometimes prevent Standard GANs on CelebA from learning initially, at which point we restart the algorithm, and once it starts learning then the run is successful. This is practically very easy to detect and provides minimal trouble, so Figure 3 focuses on successful runs. We predict this defect will be rectified with a more stable loss function, such as WGAN-GP, or with different $h$'s and $\ell$'s.

We also examine the effect of multiple generator updates compared to discriminator updates. More specifically, in RWP since we update the generator multiple times before updating the discriminator, then it is worth examining the effect of not using the regularization. We see in Figure 4 that even using the most stable GAN type out of the three – WGAN-GP – if we omit regularization then the FID has high variance and even tends to rise in the end. But with RWP, the FID converges with more stability and achieves a lower FID.

Samples from the models are provided in Appendix E. We also performed latent space walks (Radford et al., 2015) to show RWP regularization does not cause the GAN to memorize. For details see Appendix F.

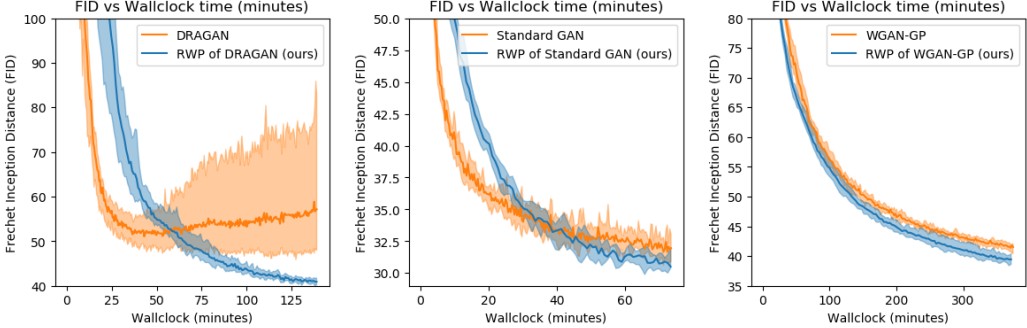

Figure 2: The effect of using RWP regularization, on the CIFAR-10 dataset. The experiments are averaged over 5 runs. The bold lines are the average, and the enveloping lines are the minimum and maximum. From the three graphs, we see that using the easy-to-implement RWP regularization improves speed as measured by wallclock time, and it also is able to achieve a lower FID.

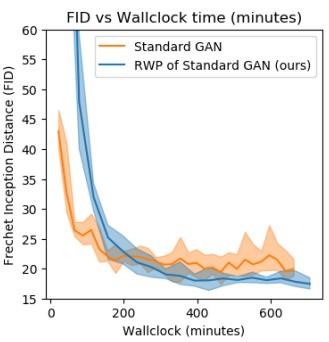

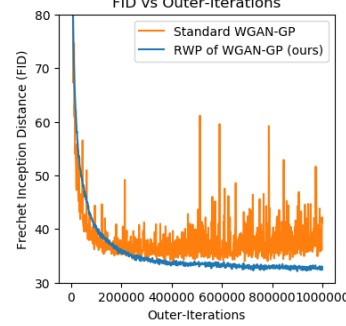

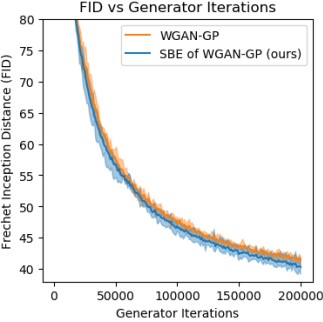

Figure 3: The effect of Relaxed Wasserstein Proximal (RWP) regularization on Standard GANs, on the CelebA dataset. The experiment was averaged over 5 runs. The bold lines are the average, and the enveloping lines are the minium and maximum. Here we see RWP regularization improves the speed (via wallclock time), and achieves a lower FID. We note multiple generator iterations might cause initial learning to fail, but once it starts then it remains successful. This is practically easy to detect, so we show successful runs.

Figure 4: An experiment demonstrating the effect of performing 10 generator iterations per outer-iteration with and without RWP, where an outer-iteration is a single loop of: a number of discriminator iterations, then a number of generator iterations. This experiment goes to 1,000,000 outer-iterations to show long-term behavior. With RWP regularization we obtain convergence, as well as lower FID. Without RWP, the training is highly variable and the FID is even on a rising trend in the end.

Figure 5: The effect of the Semi-Backward Euler (SBE) method, on the CIFAR-10 dataset. As we observe, the training is comparable to the standard way of training using the WGAN-GP loss. The experiment was averaged over 5 runs. The bold lines is the average, and the enveloping lines are the minimum and maximum.

### 3.2 RESULTS OF SEMI-BACKWARD EULER METHOD

The training of Semi-Backward Euler (SBE) is a more complicated. Here we attempt to approximate three functions: the usual discriminator and generator, and the potential function $\Phi_p$. The algorithm and particular hyperparameter settings are presented in the appendix in Section G. We present our attempts at optimizing over the three networks in Figure 5. Since both the standard WGAN-GP and the SBE on WGAN-GP had the same generator iterations, then we align according to this. As we see, the Semi-Backward Euler method is comparable to norm WGAN-GP. We leave deeper investigation of the Semi-Backward Euler method for future work.

## 4 RELATED WORKS

In the literature, many different aspects of optimal transport have been applied into machine learning and GANs.

1. *Loss function.* Many studies apply the Wasserstein distance as the loss function. There are mainly two reasons for using the Wasserstein loss function (Frogner et al., 2015; Montavon et al., 2016). On the one hand, the Wasserstein distance is a statistical distance depending on the metric of the sample space. So it introduces a statistical estimator, named the the minimal Wasserstein estimator (Bassetti et al., 2006), depending on the geometry of the data. On the other hand, the Wasserstein distance is useful for comparing probability distributions supported on lower dimensional sets. This is often intractable for other divergence functions. In GANs, these properties have been leveraged in Wasserstein GAN (Arjovsky et al., 2017). In this case, the loss function is chosen as the Wasserstein-1 distance function. In its computations, the discriminator, also called the Kantorovich dual variable, needs to satisfy the 1-Lipschitz condition. Many studies work on the regularization of the discriminator in order to satisfy this condition (Gulrajani et al., 2017b; Petzka et al., 2017).

2. *Gradient flows in full probability set.* The Wasserstein-2 metric provides a metric tensor structure (Lott, 2007; Otto, 2001; Li, 2018), under which the probability space forms an infinite dimensional Riemannian manifold, named the density manifold (Lafferty, 1988). The gradient flow in the density manifold links with many transport-related partial differential equations (Villani, 2009; Nelson, 1985). A famous example is that the Fokker-Planck equation, the probability transition equation of Langevin dynamics, is the gradient flow of the KL divergence function. In this perspective, two angles have been developed in the learning communities. Firstly, many groups try to leverage the gradient flow structure in probability space supported on the parameter space. They study the stochastic gradient descent by the transition equation in the probability over parameters (Mei et al., 2018). Secondly, many nonparametric models have been studied, such as the Stein gradient descent method (Liu, 2017). It can be viewed as the generalization of Wasserstein gradient flow. In addition, Frogner & Poggio (2018) consider an approximate inference method for computing Wasserstein gradient flow in full probability set. Here an approximation towards Kantorovich dual variables is introduced.

3. *Gradient flow constrained on parameter space.* The Wasserstein structure can also be constrained on parameter space. Carlen & Gangbo (2003) studied the constrained Wasserstein gradient with fixed mean and variance. Here the density subset is still infinite dimensional. Many approaches also focus on Gaussian families or elliptical distributions (Takatsu, 2011). The Wasserstein gradient flow in Gaussian family has been studied by Malagò et al. (2018).

Compared to previous works, our approach applies the Wasserstein gradient to work on general implicit generative models.

## 5 DISCUSSION

In this work, we apply the constrained Wasserstein gradient and its relaxations on implicit generative models. Whereas much work has focused on regularizing the discriminator, in this work we focus on regularizing the generator. For Wasserstein GAN (with gradient penalty), we compute the Wasserstein-2 gradient flow of Wasserstein-1 distance on parameter space. Experimentally, the proposed method allows us to obtain a better minimizer in the sense of FID, with faster convergence speeds in wall-clock time.

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

APPENDIX

## A  REVIEW OF WASSERSTEIN STATISTICAL MANIFOLD

In the full probability set, we consider a metric function $W_2 \colon \mathcal{P}_2(\mathbb{R}^n) \times \mathcal{P}_2(\mathbb{R}^n) \to \mathbb{R}_+$,

$$
W_2(\rho_0, \rho_1)^2 = \inf_{\Phi_t} \left\{ \int_0^1 \int_{\mathbb{R}^n} \|\nabla \Phi(t, x)\|^2 \rho(t, x) dx dt \colon \right.
$$
$$
\left. \partial_t \rho(t, x) + \nabla \cdot (\rho(t, x) \nabla \Phi(t, x)) = 0, \ \rho(0, x) = \rho_0(x), \ \rho(1, x) = \rho_1(x) \right\},
$$
$$(5)$$

where the infimum is taken among all feasible Borel potential functions $\Phi \colon [0, 1] \times \mathbb{R}^n \to \mathbb{R}$ and continuous density path $\rho \colon [0, 1] \times \mathbb{R}^n \to \mathbb{R}_+$ satisfying the continuity equation.

The variational formulation in equation 5 introduces a Riemannian structure in density space. Consider the set of smooth and strictly positive probability densities

$$
\mathcal{P}_+ = \left\{ \rho \in C^\infty(\mathbb{R}^n) \colon \rho(x) > 0, \ \int_{\mathbb{R}^n} \rho(x) dx = 1 \right\} \subset \mathcal{P}_2(\mathbb{R}^n).
$$

Denote $\mathcal{F} := C^\infty(\mathbb{R}^n)$ the set of smooth real valued functions. The tangent space of $\mathcal{P}_+$ is given by

$$
T_\rho \mathcal{P}_+ = \left\{ \sigma \in \mathcal{F} \colon \int_{\mathbb{R}^n} \sigma(x) dx = 0 \right\}.
$$

Given $\Phi \in \mathcal{F}$ and $\rho \in \mathcal{P}_+$, define

$$
V_\Phi(x) := -\nabla \cdot (\rho(x) \nabla \Phi(x)).
$$

Thus $V_\Phi \in T_\rho \mathcal{P}_+$. The elliptic operator $\nabla \cdot (\rho \nabla)$ identifies the function $\Phi$ modulo additive constants with the tangent vector $V_\Phi$ of the space of densities.

Given $\rho \in \mathcal{P}_+$, $\sigma_i \in T_\rho \mathcal{P}_+$, $i = 1, 2$, define

$$
g_\rho^W(\sigma_1, \sigma_2) = \int_{\mathbb{R}^n} (\nabla \Phi_1(x), \nabla \Phi_2(x)) \rho(x) dx,
$$

where $\Phi_i(x) \in \mathcal{F}/\mathbb{R}$, such that $-\nabla \cdot (\rho \nabla \Phi_i) = \sigma_i$.

The inner product $g^W$ endows $\mathcal{P}_+$ with a Riemannian metric tensor. In other words, the variational problem equation 5 is a geometric action energy in $(\mathcal{P}_+, g^W)$.

Given a loss function $F \colon \mathcal{P}_+ \to \mathbb{R}$, the Wasserstein gradient operator in $(\mathcal{P}_+, g^W)$ is given as follows.

$$
\operatorname{grad}_W F(\rho) = -\nabla \cdot (\rho \nabla \frac{\delta}{\delta \rho(x)} F(\rho)).
$$

Thus the gradient flow satisfies

$$
\frac{\partial \rho}{\partial t} = -\operatorname{grad}_W F(\rho) = \nabla \cdot (\rho \nabla \frac{\delta}{\delta \rho(x)} F(\rho)).
$$

More analytical results on the Wasserstein-2 gradient flow are provided in Ambrosio et al. (2005).

We next consider Wasserstein-2 metric and gradient operator constrained on statistical models. A statistical model is defined by a triplet $(\Theta, \mathbb{R}^n, \rho)$. For simple presentation of paper, we assume $\Theta \subset \mathbb{R}^d$ and $\rho \colon \Theta \to \mathcal{P}(\mathbb{R}^n)$ is a parameterization function. In this case, $\rho(\Theta) \subset \mathcal{P}(\mathbb{R}^n)$. We assume that the parameterization map $\rho$ is locally injective and under suitable regularities. We define a Riemannian metric $g$ on $\rho(\Theta)$ by pulling back the Wasserstein-2 metric tensor $g^W$.

**Definition 6 (Wasserstein statistical manifold)** *Given* $\theta \in \Theta$ *and* $\sigma_i \in T_\theta \Theta$, *$i = 1, 2$, we define*

$$
g_\theta(\sigma_1, \sigma_2) = \int_{\mathbb{R}^n} \nabla \Phi_1(x) \nabla \Phi_2(x) \rho(\theta, x) dx,
$$

*where*

$$
-\nabla \cdot (\rho(\theta, x) \nabla \Phi_i(x)) = (\nabla_\theta \rho(\theta, x), \sigma_i).
$$

*Here* $\nabla_\theta \rho = (\frac{\partial}{\partial \theta_i} \rho(\theta, x))_{i=1}^d \in \mathbb{R}^d$ *and* $(\cdot, \cdot)$ *is an Euclidean inner product in* $\mathbb{R}^d$.

In particular, we denote
$$g_\theta(\sigma, \sigma) = \sigma^\mathsf{T} G(\theta)\sigma,$$
where $G(\theta) = (G(\theta)_{ij})_{1 \le i,j \le d} \in \mathbb{R}^{d \times d}$ is the associated metric tensor defined in Theorem 2.

Here we assume that $G(\theta)$ is smooth and positive definite, so that $(\Theta, g_\theta)$ forms a smooth Riemannian manifold. In this case, Theorem 2 studies the constrained Wassertein gradient operator in parameter space.

## B  PROOFS OF WASSERSTEIN NATURAL GRADIENT

*Proof of Theorem 1* The distance $d_W$ can be written into the action function in Wasserstein statistical manifold. In other words, consider
$$d_W(\theta_0, \theta_1)^2 = \inf \left\{ \int_0^1 \dot\theta(t)^\mathsf{T} G(\theta(t))\dot\theta(t) \colon \theta(0) = \theta_0,\ \theta(1) = \theta_1 \right\}$$
where the infimum is taken over $\theta(t) \in C^1([0,1], \Theta)$. Following the definition of metric tensor in definition 6, we have
$$\dot\theta(t)^\mathsf{T} G(\theta(t))\dot\theta(t) = \int_{\mathbb{R}^n} (\nabla\Phi(t,x))^2 \rho(\theta(t), x) dx,$$
with $\Phi(t,x)$ satisfying
$$\partial_t \rho(\theta(t), x) = \nabla_\theta \rho(\theta(t), x)\dot\theta(t) = -\nabla(\rho(\theta(t), x)\nabla\Phi(t,x)).$$
We finish the proof.

*Proof of Theorem 2* The gradient operator on a Riemannian manifold $(\Theta, g_\theta)$ is defined as follows.

For any $\sigma \in T_\theta\Theta$, then the Riemannian gradient $\nabla_\theta^W F(\theta) \in T_\theta\Theta$ satisfies
$$g_\theta(\sigma, \nabla_\theta^W F(\theta)) = (\nabla_\theta F(\theta), \sigma).$$
In other words,
$$\dot\theta^\mathsf{T} G(\theta)\nabla_\theta^W F(\theta) = \nabla_\theta F(\theta)^\mathsf{T}\sigma.$$
Since $\theta \subset \mathbb{R}^d$ and $G(\theta)$ is positive definite, then
$$\nabla_\theta^W F(\theta) = G(\theta)^{-1}\nabla_\theta F(\theta).$$

*Proof of Proposition 3.* We next present the derivation of the proposed semi-backward method.

**Claim:** Denote $\|\theta - \theta^k\| = h$, then
$$(\theta^k - \theta)^\mathsf{T} G(\theta^k)(\theta^k - \theta) = d_W(\theta, \theta^k)^2 + O(h^2), \tag{6}$$
and
$$\frac{1}{2}(\theta^k - \theta)^\mathsf{T} G(\theta^k)(\theta^k - \theta) + O(h^2) = \sup_\Phi \int_{\mathbb{R}^n} \Phi(x)(\rho(\theta, x) - \rho(\theta^k, x)) - \frac{1}{2}\|\nabla\Phi(x)\|^2 \rho(\theta^k, x) dx. \tag{7}$$

*Proof of Claim.* We next prove the claim. Denote the geodesic path $\theta^*(t)$, $t \in [0,1]$, with $\theta^*(0) = \theta$, $\theta^*(1) = \theta^k$, s.t.
$$d_W(\theta, \theta^k)^2 = \int_0^1 (\frac{d}{dt}\theta^*(t))^\mathsf{T} G(\theta^*(t))\frac{d}{dt}\theta^*(t) dt.$$
We reparameterize the time of $\theta^*(t)$ into the time interval $[0, h]$. Denote $\tau = ht$ and $\theta(\tau) = \theta^*(ht)$. Thus $\theta(\tau) = \theta^k + \frac{\theta - \theta^k}{h}\tau + O(\tau^2)$ and $\frac{d}{d\tau}\theta(\tau) = \frac{\theta - \theta^k}{h} + O(\tau)$,
$$\begin{aligned}
d_W(\theta, \theta^k)^2 &= h \int_0^h \frac{d}{d\tau}\theta(\tau)^\mathsf{T} G(\theta(\tau))\frac{d}{d\tau}\theta(\tau) d\tau \\
&= h \int_0^h (\frac{\theta - \theta^k}{h} + O(h))^\mathsf{T} G(\theta^k + O(h))(\frac{\theta - \theta^k}{h} + O(h)) d\tau \\
&= (\theta - \theta^k)^\mathsf{T} G(\theta^k)(\theta - \theta^k) + O(h^2),
\end{aligned}$$

which proves equation 6.

We next prove equation 7. On the L.H.S. of equation 7,

$$\nabla_\theta \rho(\theta^k, x)(\theta - \theta^k) = \rho(\theta, x) - \rho(\theta^k, x) + O(h).$$

From the definition of $G(\theta)$,

$$\frac{1}{2}(\theta - \theta^k)^\mathsf{T} G(\theta^k)(\theta - \theta^k) = \frac{1}{2}\int_{\mathbb{R}^n} (\nabla \Phi(x))^2 \rho(\theta^k, x) dx,$$

where

$$-\nabla \cdot (\rho(\theta^k, x)\nabla \Phi(x)) = \nabla_\theta \rho(\theta^k, x)(\theta - \theta^k) = \rho(\theta^k, x) + O(h).$$

On the R.H.S. of equation 7, the maximizer $\Phi^*$ satisfies

$$\rho(\theta, x) - \rho(\theta^k, x) + \nabla \cdot (\rho(\theta^k, x)\nabla \Phi^*(x)) = 0. \tag{8}$$

Applying equation 8 into the R.H.S. of equation 7, we have

$$\int_{\mathbb{R}^n} \Phi^*(x)(\rho(\theta, x) - \rho(\theta^k, x)) - \frac{1}{2}\|\nabla \Phi^*(x)\|^2 \rho(\theta^k, x) dx$$

$$= \int_{\mathbb{R}^n} \Phi^*(x)[-\nabla \cdot (\rho(\theta^k, x)\nabla \Phi^*(x)] - \frac{1}{2}\nabla \Phi^*(x)\rho(\theta^k, x) dx$$

$$= \int_{\mathbb{R}^n} \|\nabla \Phi^*(x)\|^2 \rho(\theta^k, x) - \frac{1}{2}\|\nabla \Phi^*(x)\|^2 \rho(\theta^k, x) dx$$

$$= \frac{1}{2}\int_{\mathbb{R}^n} \|\nabla \Phi^*(x)\|^2 \rho(\theta^k, x) dx.$$

Comparing the L.H.S. and R.H.S. of equation 7, we prove the claim.

From the claim,

$$\theta^{k+1} = \arg\min_{\theta \in \Theta} F(\theta) + \frac{1}{h}\frac{d_W(\theta, \theta^k)^2}{2}$$

$$= \arg\min_{\theta \in \Theta} F(\theta) + \frac{1}{2h}(\theta^k - \theta)^\mathsf{T} G(\theta^k)(\theta^k - \theta) + O(h)$$

$$= \arg\min_{\theta \in \Theta} F(\theta) + \frac{1}{h}\sup_\Phi \int_{\mathbb{R}^n} \Phi(x)(\rho(\theta, x) - \rho(\theta^k, x)) - \frac{1}{2}\|\nabla \Phi(x)\|^2 \rho(\theta^k, x) dx + O(h).$$

Thus we derive a consistent numerical method in time, known as the Semi-backward method:

$$\theta^{k+1} = \theta^k - hG(\theta^k)^{-1}\nabla_\theta F(\theta^{k+1}).$$

*Proof of Proposition 4.* This result is proven in Li & Osher (2018). We present it here for the completion of paper. The implicit model is given by the following push-forward relation. Denote $g_\theta \# p(z) = \rho(\theta, x)$, i.e.,

$$\int_{\mathbb{R}^m} f(g(\theta, z))p(z)dz = \int_{\mathbb{R}^n} f(x)\rho(\theta, x)dx, \quad \text{for any } f \in C_c^\infty(\mathbb{R}^n). \tag{9}$$

Given the gradient constraint

$$\frac{d}{dt}g(\theta(t), z) = \nabla \Phi(t, g(\theta(t), z)),$$

we shall show that the probability density transition equation of $g(\theta(t), z)$ satisfies the *constrained continuity equation*

$$\frac{\partial}{\partial t}\rho(\theta(t), x) + \nabla \cdot (\rho(\theta(t), x)\nabla \Phi(t, x)) = 0, \tag{10}$$

and

$$\mathbb{E}_{Z \sim p(z)}\|\frac{d}{dt}g(\theta(t), Z)\|^2 = \int_{\mathbb{R}^n} \|\nabla \Phi(t, x))\|^2 \rho(\theta(t), x)dx. \tag{11}$$

On the one hand, consider $f \in C_c^\infty(\mathbb{R}^n)$, then

$$
\begin{aligned}
\frac{d}{dt}\mathbb{E}_{Z\sim p(z)}f(g(\theta(t), Z)) &= \frac{d}{dt}\int_{\mathbb{R}^m} f(g(\theta(t), z))p(z)dz \\
&= \frac{d}{dt}\int_{\mathbb{R}^n} f(x)\rho(\theta(t), x)dx \\
&= \int_{\mathbb{R}^n} f(x)\frac{\partial}{\partial t}\rho(\theta(t), x)dx,
\end{aligned}
\tag{12}
$$

where the second equality holds from the push forward relation in equation 9.

On the other hand, consider

$$
\begin{aligned}
\frac{d}{dt}\mathbb{E}_{Z\sim p(z)}f(g(\theta(t), Z)) &= \lim_{\Delta t \to 0}\mathbb{E}_{Z\sim p(z)}\frac{f(g(\theta(t+\Delta t), Z)) - f(g(\theta(t), Z))}{\Delta t} \\
&= \lim_{\Delta t \to 0}\int_{\mathbb{R}^m}\frac{f(g(\theta(t+\Delta t), z)) - f(g(\theta(t), z))}{\Delta t}p(z)dz \\
&= \int_{\mathbb{R}^m}\nabla f(g(\theta(t), z))\frac{d}{dt}g(\theta(t), z)p(z)dz \\
&= \int_{\mathbb{R}^m}\nabla f(g(\theta(t), z))\nabla\Phi(t, g(\theta(t), z))p(z)dz \\
&= \int_{\mathbb{R}^n}\nabla f(x)\nabla\Phi(t, x)\rho(\theta(t), x)dx \\
&= -\int_{\mathbb{R}^n} f(x)\nabla\cdot(\nabla\Phi(t, x)\rho(\theta(t), x))dx,
\end{aligned}
\tag{13}
$$

where $\nabla$, $\nabla\cdot$ are gradient and divergence operators w.r.t. $x \in \mathbb{R}^n$. The second to last equality holds from the push forward relation equation 9, and the last equality holds using the integration by parts w.r.t. $x$. Since $equation$ 12 equals $equation$ 13 for any $f \in C_c^\infty(\mathbb{R}^n)$, we prove equation 10.

In addition, by the definition of the push forward operator equation 9, we have

$$
\begin{aligned}
\mathbb{E}_{Z\sim p(z)}\|\frac{d}{dt}g(\theta(t), Z)\|^2 &= \int_{\mathbb{R}^n}\|\nabla\Phi(t, g(\theta(t), z))\|^2 p(z)dz \\
&= \int_{\mathbb{R}^n}\|\nabla\Phi(t, x)\|^2\rho(\theta(t), x)dx.
\end{aligned}
$$

Thus we prove equation 11.

*Proof of Proposition 5.* This example allows us to compute the proximal operator explicitly. On the one hand, we compute the Wasserstein proximal operator explicitly:

$$
\begin{aligned}
\theta_W^{k+1} = (a_W^{k+1}, b_W^{k+1}) &= \arg\min_\theta F_{W_1}(\theta) + \frac{1}{2h}d_W(\theta, \theta^k)^2 \\
&= \arg\min_{(a,b)} \alpha|a - a^*| + (1-\alpha)|b - b^*| + \frac{1}{2h}(\alpha|a - a^k|^2 + (1-\alpha)|b - b^k|).
\end{aligned}
$$

I.e.,

$$
a_{k+1}^W = \arg\min_a |a - a^*| + \frac{1}{2h}|a - a^k|^2, \quad b_{k+1}^W = \arg\min_b |b - b^*| + \frac{1}{2h}|b - b^k|^2.
$$

Here

$$
a_W^{k+1} = \text{shrink}_{a^*}(a^k, h) = \begin{cases} a^k - h & \text{if } a^k > a^* + h; \\ a^k + h & \text{if } a^k < a^* - h; \\ a^* & \text{otherwise.} \end{cases}
$$

Similarly, $b_W^{k+1} = \text{shrink}_{b*}(b^k, h)$.

On the other hand, we calculate the Euclidean proximal operator explicitly:

$$
\begin{aligned}
\theta_E^{k+1} = (a_E^{k+1}, b_E^{k+1}) &= \arg\min_\theta F_{W_1}(\theta) + \frac{1}{2h}d_E(\theta, \theta^k)^2 \\
&= \arg\min_{(a,b)} \alpha|a - a^*| + (1-\alpha)|b - b^*| + \frac{1}{2h}(|a - a^k|^2 + |b - b^k|^2).
\end{aligned}
$$

I.e.,

$$a_E^{k+1} = \arg\min_a \alpha|a - a^*| + \frac{1}{2h}|a - a^k|^2, \quad b_E^{k+1} = \arg\min_b (1 - \alpha)|b - b^*| + \frac{1}{2h}|b - b^k|^2.$$

Here

$$a_E^{k+1} = \text{shrink}_{a^*}(a^k, \alpha h) = \begin{cases} a^k - \alpha h & \text{if } a^k > a^* + \alpha h; \\ a^k + \alpha h & \text{if } a^k < a^* - \alpha h; \\ a^* & \text{otherwise.} \end{cases}$$

Similarly, $b_E^{k+1} = \text{shrink}_{b*}(b^k, (1 - \alpha)h)$.

Here we only need to check that for all possible cases, $F_{W_1}(\theta_E^{k+1}) > F_{W_1}(\theta_W^{k+1})$. If $a^k > a^* + h$ and $b^k > b^* + h$, then

$$F_{W_1}(\theta_W^{k+1}) = \alpha[(a^k - a^* - h) + \frac{h}{2}] + (1 - \alpha)[(b^k - b^* - h) + \frac{h}{2}]$$
$$= \alpha(a^k - a^*) + (1 - \alpha)(b^k - b^*) - \frac{h}{2},$$

and

$$F_{W_1}(\theta_E^{k+1}) = \alpha[(a^k - a^* - \alpha h)] + \frac{(\alpha h)^2}{2h} + (1 - \alpha)[(b^k - b^* - \alpha h)] + \frac{(1 - \alpha)^2 h^2}{2h}$$
$$= \alpha(a^k - a^*) + (1 - \alpha)(b^k - b^*) - \frac{h}{2}[\alpha^2 + (1 - \alpha)^2].$$

Since $\alpha \in [0, 1]$, then $\alpha^2 + (1 - \alpha)^2 \leq [\alpha + (1 - \alpha)]^2 = 1$, then $F_{W_1}(\theta_W^{k+1}) \leq F_{W_1}(\theta_E^{k+1})$. In other cases, the proof follows similarly. We finish the proof.

## C  HYPERPARAMETERS FOR RELAXED WASSERSTEIN PROXIMAL EXPERIMENTS

The following hyperparameter settings for the Relaxed Wasserstein Proximal experiments in Section 3.1 are:

- A batch size of 64 for all experiments.
- For CIFAR-10 with WGAN-GP: The Adam optimizer with learning rate 0.0001, $\beta_1 = 0.5$, and $\beta_2 = 0.9$ for both the generator and discriminator. We used a latent space dimension of 128, $h = 0.1$, and $\ell = 10$ generator iterations.
- For CIFAR-10 with Standard and DRAGAN: The Adam optimizer with learning rate 0.0002, $\beta_1 = 0.1$, and $\beta_2 = 0.999$ for both the generator and discriminator. We used a latent space dimension of 100, $h = 0.2$, and $\ell = 5$ generator iterations.
- For aligned and cropped CelebA with Standard: The Adam optimizer with learning rate 0.0002, $\beta_1 = 0.5$, and $\beta_2 = 0.999$ for both the generator and discriminator. We used a latent space dimension of 100, $h = 0.2$, and $\ell = 5$ generator iterations.

## D  A PRACTICAL DESCRIPTION OF THE RELAXED WASSERSTEIN PROXIMAL

As mentioned in Section 3.1, the Relaxed Wasserstein Proximal is meant to be an easy-to-implement, drop-in regularization. For instructional purposes, we take a specific example to showcase the algorithm: Relaxed Wasserstein Proximal on Standard GANs (with non-saturating gradient for the generator):

- Given:
  - A generator $g_\theta$, and discriminator $D_\omega$,
  - The distance function $F_\omega(g_\theta) = \mathbb{E}_{x \sim \text{real}}[\log(D_\omega(x))] - \mathbb{E}_{z \sim \mathcal{N}(0,1)}[\log(1 - D_\omega(g_\theta(z)))]$,

- Choice of optimizers, $\text{Adam}_\omega$ and $\text{Adam}_\theta$,

- Proximal step-sizes $h$, and generator iterations $\ell$, and

- Batch size $B$.

Then the algorithm follows:

1. Sample real data $\{x_i\}_{i=1}^B$, and latent data $\{z_i\}_{i=1}^B$.

2. Update the discriminator:

$$\omega^k \leftarrow \text{Adam}_\omega \left( -\frac{1}{B} \sum_{i=1}^B \log(D_\omega(x_i)) - \frac{1}{B} \sum_{i=1}^B \log(1 - D_\omega(g_\theta(z_i))) \right)$$

3. Sample latent data $\{z_i\}_{i=1}^B$

4. Perform Adam gradient descent $\ell$ number of times:

$$\theta^k \leftarrow \text{Adam}_\theta \left( -\frac{1}{B} \sum_{i=1}^B \log(D_\omega(g_\theta(z_i))) - \frac{1}{B} \sum_{i=1}^B \frac{1}{2h} \|g_\theta(z_i) - g_{\theta^{k-1}}(z_i)\|_2^2 \right),$$
$$\text{for } \ell \text{ number of times.}$$

5. Repeat the above until a chosen stopping condition (e.g. maximum number of iterations).

As one can analyze above, the only difference between the standard way of training GANs and using the Relaxed Wasserstein Proximal, are the $\|g_\theta(z_i) - g_{\theta^{k-1}}(z_i)\|_2^2$ terms and the number of generator iterations $\ell$. Note that in this paper, we call a single loop of updating a discriminator a number of times and then updating the generator a number of a time, an outer-iteration.

## E   GENERATED SAMPLES FROM THE MODEL

In Figure 6, we have samples generated from a Standard GAN with RWP regularization, trained on the CelebA dataset. The FID of these images was 17.105.

In Figure 7, we have samples generated from WGAN-GP with RWP , trained on the CIFAR-10 dataset. The FID for these images is 38.3.

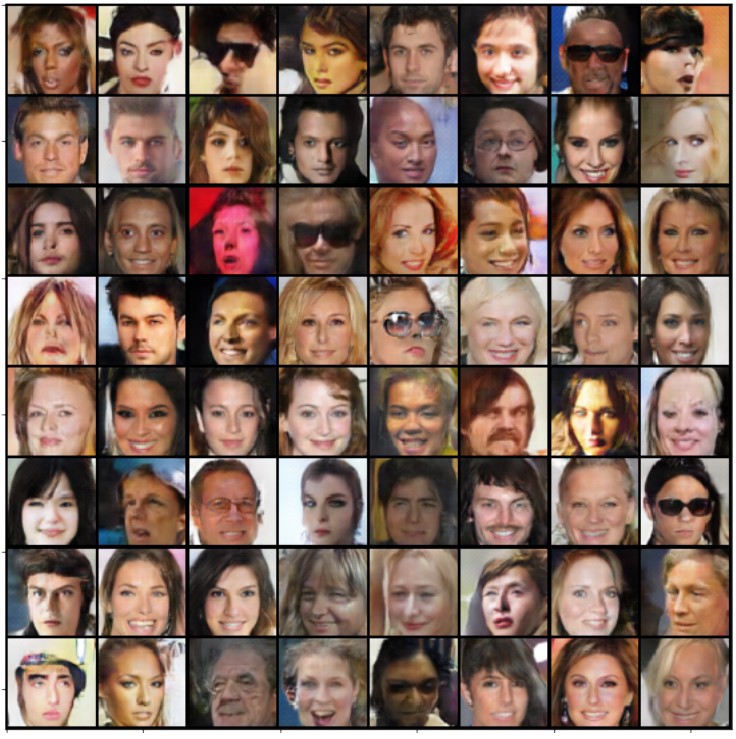

Figure 6: A sample of images generated by RWP regularization on Standard GANs, on CelebA.

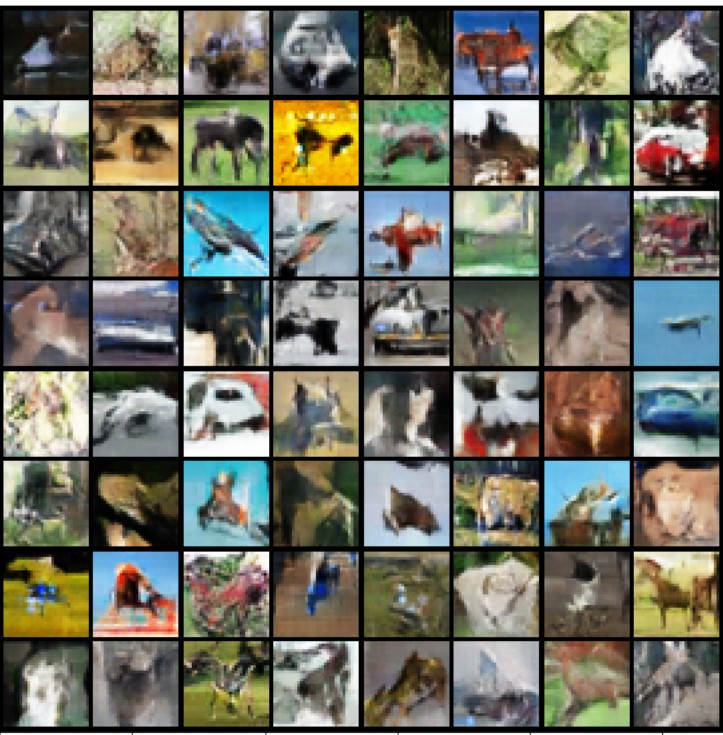

Figure 7: A sample of images generated by RWP regularization on WGAN-GP, on CIFAR-10.

## F  LATENT SPACE WALK

Radford et al. (2015) suggest that walking in the latent space could detect whether a generator was memorizing. We see in Figure 8 and Figure 9 that we have smooth transitions, so this is not the case for GANs with RWP regularization.

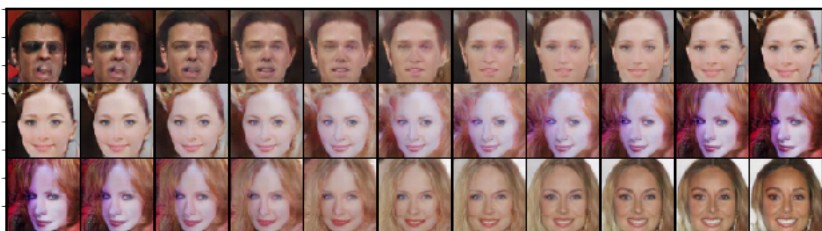

Figure 8: A latent space walk for a network with RWP regularization on Standard GANs, on CelebA. As we have smooth transitions, this shows the generator is not overfitting. The latent space walk is done by interpolating between 4 points in the latent space.

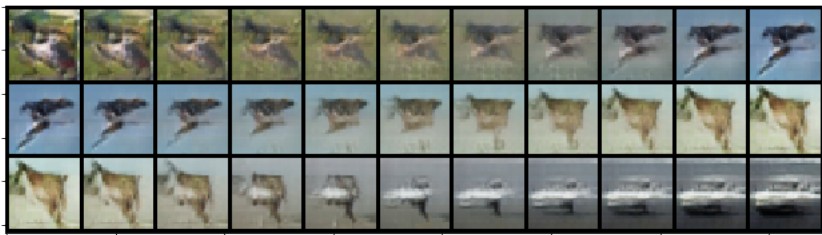

Figure 9: A latent space walk for a network with RWP regularization on WGAN-GP, on CIFAR-10. As we have smooth transitions, this shows the generator is not overfitting. The latent space walk is done by interpolating between 4 points in the latent space.

## G  ALGORITHM AND PARTICULAR HYPERPARAMETERS FOR THE SEMI-BACKWARD EULER METHOD

The specific hyperparameter settings used for the Semi-Backward Euler (SBE) on WGAN-GP, trained on CIFAR-10, are:

- A batch size of 64.
- The DCGAN architecture for the discriminator and generator. A one-hidden-layer fully-connected network (a.k.a. MLP) for the potential $\Phi_p$. We also used layer-normalization (Lei Ba et al. (2016)) for each layer.
- We used the Adam optimizer with learning rate $0.0002$, $\beta_1 = 0.1$, and $\beta_2 = 0.999$ for both the generator, discriminator, and potential $\Phi_p$. We used a latent space dimension of 100, and $h = 0.2$.
- Every outer-iteration loop, we updated the discriminator 5 times (as suggested in WGAN-GP), the generator once, and the potential 5 times. Note an outer-iteration is defined as one loop of: updating the discriminator a number of times, updating the potential a number of times, and updating the generator a number of times.

---

**Algorithm 2** Semi-backward Euler method, where $F_\omega$ is a parameterized function to minimize.

---

**Require:** $F_\omega$, a parameterized function to minimize (e.g. Wasserstein-1 with a parameterized discriminator). $g_\theta$ the generator. $\Phi_p$ the potential.
**Require:** $h$ the proximal step-size, $m$ the batch size.
**Require:** Optimizer$_{F_\omega}$, Optimizer$_{g_\theta}$, and Optimizer$_{\Phi_p}$
**Require:** The number of generator iterations and p iterations to do per update.
1: **for** $k = 0$ **to** max iterations **do**
2:     Sample real data $\{x_i\}_{i=1}^B$ and latent data $\{z_i\}_{i=1}^B$.
3:     $\omega^k \leftarrow$ Optimizer$_{F_\omega}\left(\frac{1}{B}\sum_{i=1}^B F_\omega(g_\theta(z_i))\right)$
4:     **for** $s = 0$ **to** phi iterations **do**
5:         Sample latent data $\{z_i\}_{i=1}^B$
6:         $p^k \leftarrow$ Optimizer$_{\Phi_p}\left(\frac{1}{h}\frac{1}{B}\sum_{i=1}^B \Phi_p(g_\theta(z_i)) - \Phi_p(g_{\theta^{k-1}}(z_i)) - \frac{1}{2}\nabla\Phi_p(g_{\theta^{k-1}}(z_i))\right)$
7:     **end for**
8:     **for** $\ell = 0$ **to** generator iterations **do**
9:         Sample latent data $\{z_i\}_{i=1}^B$
10:         $\theta^k \leftarrow$ Optimizer$_{g_\theta}\left(\frac{1}{B}\sum_{i=1}^B F_\omega(g_\theta(z_i)) + \frac{1}{h}\left(\Phi_p(g_\theta(z_i)) - \Phi_p(g_{\theta^{k-1}}(z_i)) - \frac{1}{2}\nabla\Phi_p(g_{\theta^{k-1}}(z_i))\right)\right)$
11:     **end for**
12: **end for**

---

