# OpenReview forum: "Wasserstein proximal of GANs"
_ICLR.cc/2019/Conference_

### Official Review · AnonReviewer1 · 2018-10-31
**An interesting paper but need more work in the context of GANs**

**Rating:** 4
**Confidence:** 3

**Review:**

The paper intends to utilize natural gradient induced by Wasserstein-2 distance to train the generator in GAN. Starting from the dynamical formulation of optimal transport, the authors propose the Wasserstein proximal operator as a regularization, which is simple in form and fast to compute. The proximal operator is added to training the generator, unlike most other regularizations that focus on the discriminator. This is an interesting direction.

The motivation is clear but by so many steps of approximation and relaxation, the authors didn’t address what is the final regularization actually corresponding to? Personally I am not convinced that theoretically the proposed training method is better than the standard SGD. The illustration example in the paper is not very helpful as it didn’t show how the proposed proximal operator works. The proximal operator serves as a regularization and it introduces some error, I would like to know how does this carry over to the whole training procedure.

In GAN, the optimal discriminator depends on the current generator. Many approaches to GAN training (i.e. WGAN-GP) advocates to update the generator once in every “outer-iteration”. I am not sure how the proposed approach fit in those training schemes.

In the simulation, the difference is not very significant, especially in FID vs iteration number. This could be due to parameter tuning in standard WGAN-GP. I encourage more simulation studies and take more GAN structures into consideration.

Lastly, the stability mentioned in the paper lacks a formal definition. Is it the variance of the curves? Is it how robust the model is against outer iterations?

---

> ### Author Response · Authors · 2018-11-17
> **Response to AnonReviewer1**
>
> Thank you for your comments!
>
> Summary of the response: RWP is an easy to implement method with a rigorous mathematical motivation. It can be interpreted as a stochastic version of SBE. Proximal is not an approximation. The proximal update is defined implicitly as the optimizer of a subproblem, which is why we have several generator updates per outer iteration. Experiments show that our method is faster and more stable than state-of-the-art methods.
>
> In terms of what RWP corresponds to:
> RWP can be seen as a stochastic version of the Semi-Backward Euler approach. To see this, note that if we write the loss function as Loss(G) + E[Phi(G_theta) – Phi(G_theta_k-1) - (½) grad(Phi)(G_theta_k-1)^2], and constrain Phi(x) = a^Tx + b, and to one sample, then solving for the parameters a and b (setting the gradient equal to zero), we obtain RWP for each single patch. We will make this clearer in the paper.
>
> In terms of approximations in the proximal method:
> The proximal operator is a way of formulating a sequential optimization method. This is not an approximation. In the proximal formulation, each parameter update is expressed implicitly as the minimizer of the original objective function plus a penalty on the size of the step. In the case of the Wasserstein proximal, the size of the step is measured by means of the Wasserstein metric.
>
> Our motivation for using the proximal formulation of the Wasserstein gradient is twofold:
> 1) it allows us to compute the parameter updates efficiently.
> 2) it is stable and can handle non-smooth loss functions (the Wasserstein-1 loss in WGANs is non-smooth).
>
> By working with the constrained Wasserstein distance, we can obtain a tractable (finite dimensional) version of the proximal. The SBE method is obtained by second-order expansion of the proximal penalty term d(theta,theta_k). This is an approximation as much as any finite step size gradient method is an approximation of the infinitesimal gradient flow.
>
> We do introduce an approximation when we derive SBE (see above) and RWP (specifically, we drop the gradient constraint on \Phi). This serves the purpose of obtaining a scheme that is as simple as possible. Many practical algorithms follow the principle of simplicity and often work more robustly than other theoretically more accurate methods. Our experiments are in line with this idea and show that RWP offers a robust easy to implement optimization method, which can obtain faster and more stable convergence in state-of-the-art GANs.
>
> In terms of the outer iterations:
> Yes, many approaches to GAN training advocate updating the generator once every outer iteration. However, note that our inner iteration is solving for the proximal update. The proximal step is defined implicitly as the minimizer of a subproblem. In order to obtain the update, we run a short optimization loop, which is the sequence of generator updates per outer iteration. We have found that updating the generator multiple times in every outer iteration leads to better results. We think that this is a valuable observation.
>
> In the experiments:
> We do believe the results are significant, as we are comparing wallclock time. This can be especially seen in DRAGAN. Even in CIFAR10 with WGAN-GP, if we look at the average lines we observe that RWP is about 20% faster at reaching the same final FID value. And it also achieves a lower FID at the same time. We agree that more hyperparameter tuning will probably give even better results.
>
> In terms of our illustration with the toy example: We are happy to add more intuitions and illustrations to convey a clearer picture of what the proximal method does in comparison to the gradient method.
>
> For the stability results:
> Yes, it is the curves, namely when we trained WGAN-GP for one million iterations with and without RWP regularization. We find the curves are less oscillatory with RWP.
>
> Thanks again for your comments.

---

### Official Review · AnonReviewer2 · 2018-11-01
**Providing an easy-to-implement drop-in regularizer framework, which may simply be viewed as a naive application of the proximal operator.**

**Rating:** 6
**Confidence:** 3

**Review:**


[summary]
This paper considers natural gradient learning in GAN learning, where the Riemannian structure induced by the Wasserstein-2 distance is employed. More concretely, the constrained Wasserstein-2 metric $d_W$, the geodesic distance on the parameter space induced by the Wasserstein-2 distance in the ambient space, is introduced (Theorem 1). The natural gradient on the parameter space with respect to the constrained Wasserstein-2 metric is then derived (Theorem 2). Since direct evaluation of $G^{-1}$ poses difficulty, the authors go on to considering a backward scheme using the proximal operator (3), yielding:
(i) The Semi-Backward Euler method is proposed via a second-order Taylor approximation of the proximal operator $d_W^2$ (Proposition 3).
(ii) From an alternative formulation for $d_W$ (Proposition 4), the authors propose dropping the gradient constraint to define a relaxed Wasserstein metric $d$, yielding a simple proximal operator given by the expected squared Euclidean distance in the sample space used as a regularizer (equation (4)). The resulting algorithm is termed the Relaxed Wasserstein Proximal (RWP) algorithm.

[pros]
The proposal provides an easy-to-implement drop-in regularizer framework, so that it can straightforwardly be combined with various generator update schemes.

[cons]
Despite all the theoretical arguments given to justify the proposal, the resulting proposal may simply be viewed as a naive application of the proximal operator.

[Quality]
See [Detailed comments] section below.

[Clarity]
This paper is basically clearly written.

[Originality]
Providing justification to the proximal operator approach in GAN learning via natural gradient with respect to the Riemannian structure seems original.

[Significance]
See [Detailed comments] section below.

[Detailed comments]
To the parameter space $\Theta\subset\mathbb{R}^d$, one can consider introducing several different Riemannian structures, including the conventional Euclidean structure and that induced by the Wasserstein-2 metric. Which Riemannian structure among all these possibilities would be natural and efficient in GAN training would not be evident, and this paper discusses this issue only in the very special single instance in Section 2.3. A more thorough argument supporting superiority of the Riemannian structure induced by the Wasserstein-2 metric would thus be needed in order to justify the proposed approach.

In relation to this, the result of comparison between WGAN-GP with and without SBE shown in Figure 5 is embarrassing to me, since it might suggest that the proposed framework aiming at performing Wasserstein natural gradient is not so efficient if combined with WGAN-GP. The natural gradient is expected to be efficient when the underlying coordinate system is non-orthonormal (Amari, 1998). Starting with the gradient descent iteration derived from the backward Euler method in (3), which is computationally hard, the argument in this paper goes on to propose two methods: the Semi-Backward Euler method via a second-order Taylor approximation to the backward Euler scheme (Proposition 3), and RWP in (4) via approximation (dropping of the gradient constraint and finite-difference approximation in the integral with respect to $t$) of an alternative simpler formulation for the Wasserstein metric (Proposition 4). These two methods involve different approximations to the Semi-Backward Euler, and one would like to know why the approximations in the latter method is better in performance than those in the former. Discussion on this point is however missing in this paper.

In Section 3, it would have been better if the performance be compared not only in terms of FID but also the loss considered (i.e., Wasserstein-1), since the latter is exactly what the algorithms are trying to optimize.

Minor points:

Page 4: The line just after equation (4) should be moved to the position following the equation giving $d(\theta_0,\theta_1)^2$.

In the reference list, the NIPS paper by Gulrajani et al. appears twice.

---

> ### Author Response · Authors · 2018-11-17
> **Response to AnonReviewer2**
>
> Thank you for your comments!
>
> Summary of the response: Proximal is not an approximation. RWP is an easy to implement method with a rigorous mathematical motivation. RWP is different from the naive proximal. Natural gradients define distances based on the actual arguments of the objective function and are natural in this sense. Our proximal Wasserstein natural gradient solves the two major bottlenecks for using natural gradients in practice, namely tractability, and stability. Experiments show that our method is faster and more stable than state-of-the-art methods.
>
> In terms of the cons:
> We agree that the end result is simple, which was also our intention. However, we do not agree that it can be viewed as a naive application of the proximal operator. A naive implementation of a proximal operator would penalize the size of the parameter update, i.e., \|theta - theta_{k-1}\|^2, but our method penalizes the expected value of the distance between generator samples, i.e., \|g_\theta(z) - g_{\theta^{k-1}}(z)\|^2.
>
> We do agree that we have many theoretical derivations, and the resulting application is simple. Indeed, in the paper, we seek for a simple and easy to implement method. We humbly argue that, even if the end result is simple, the theoretical derivations give a mathematically sound motivation, interpretation, and a basis to continue working on related methods. In particular, we formulate the Wasserstein proximal and the Wasserstein Semi-backward Euler methods, both of which can be used as a starting point for practical optimization methods for GANs.
>
> In regard to the natural geometry of parameter space:
> We agree that the natural choice of a geometry for parameter space is not obvious. However, it is natural to define an optimization method that is independent of the parametrization, which is the case for the Wasserstein gradient but is not the case for the Euclidean gradient. Natural gradients define the geometry on parameter space based on the geometry of the functions that these parameters represent. Since the objective functions that we are optimizing depend on these functions, but not on the parameter, the natural gradients are the natural choice. The Fisher natural gradient has been observed to perform better than the Euclidean gradient in numerous statistics and machine learning applications.
>
> There are two main reasons why natural gradients are not commonly used in the place of regular gradients: 1) The computation is expensive. 2) The numerics might be unstable, especially when the step size is large. In both regards, our proposed method performs well: 1) Our RWP has no significant additional cost in computation over the regular gradient. 2) Our experiments show that the method is stable and even delivers a more stable convergence than regular gradients. In summary, by using the proximal formulation of the natural Wasserstein gradient, we address the two major obstacles in using natural gradients.
>
> In Section 2.3 we use a toy example to illustrate the differences between Wasserstein and standard gradients. For this example, we show that for any fixed step size, the Wasserstein proximal is better than the Euclidean proximal, which means that the Wasserstein gradient flow is better than the Euclidean gradient flow.
>
> The SBE and RWP are two ways of implementing the Wasserstein proximal. SBE is a precise method, although the parameter update is defined implicitly and requires solving an additional optimization problem. On the other hand, RWP tries to formulate a method that is as simple as possible, even if it is not as accurate.
>
> WGANs try to optimize the Wasserstein-1 loss, but currently there are no known practical means of computing the Wasserstein-1 loss. In the literature, the performance of trained GANs has been evaluated by means of visual inspection of the samples, and, more recently, more quantitative methods have been proposed. FID is currently considered the most informative one. This is why we focus on FID. Finding good measures for the performance of a trained GAN is a pressing and active area of investigation in the context of GANs. Addressing this is in any case beyond the scope of this paper.
>
> Thanks again for your helpful comments.

---

### Official Review · AnonReviewer3 · 2018-11-04
**ultimately, I am not sure there is anything "Wasserstein" going on in this new GAN algorithm.**

**Rating:** 3
**Confidence:** 5

**Review:**

The authors propose a new GAN procedure. It's maybe easier to reverse-engineer it from the simplest of all places, that is p.16 in the appendix which makes explicit the difference between this GAN and the original one: the update in the generator is carried out l times and takes into account points generated in the previous iteration.

To get there, the authors take the following road: they exploit the celebrated Benamou-Brenier formulation of the W2 distance between probability measures, which involves integrating over a vector field parameterized in time. The W2 distance which is studied here is not exactly that corresponding to the measures associated with these two parameters, but instead an adaptation of BB to parameterized measures ("constrained"). This metric defines a Riemannian metric between two parameters, by considering the resulting vector field that solve this equation (I guess evaluated at time 0). The authors propose to use the natural gradient associated with that Riemannian metric (Theorem 2). Using exactly that natural gradient would involve solving an optimal transport problem (compute the optimal displacement field) and inverting the corresponding operator. The authors mention that, equivalently, a JKO type step could also be considered to obtain an update for \theta. The authors propose two distinct approximations, a "semi-backward Euler formulation", and, next, a simplification of the d_W, which, exploiting the fact that one of the parameterized measures is the push foward of a Gaussian, simplifies to a simpler problem (Prop. 4). That problem introduces a new type of constraint (Gradient constraint) which is yet again simplified.

In the end, the metric considered on the parameter space is fairly trivial and boils down to the r.h.s. of equation 4. It's essentially an expected squared distance between the new and the old parameter under a Gaussian prior for the encoder.  This yields back the simplification laid out in p.16.

I think the paper is head over heels. It can be caricatured as extreme obfuscation for a very simple modification of the basic GAN algorithm. Although I am *not* claiming this is the intention of the authors, and can very well believe that they found it interesting that so many successive simplifications would yield such a simple modification, I believe that a large pool of readers at ICLR will be extremely disappointed and frustrated to see all of this relatively arduous technical presentation produce such a simple result which, in essence, has absolutely nothing to do with the Wasserstein distance, nor with a "Wasserstein natural gradient".

other comments::

*** "Wasserstein-2 distance on the full density set": what do you mean exactly? that d_W(\theta_0,\theta_1) \ne W(p_{\theta_0},p_{\theta_1})? Could you elaborate where this analogy breaks down?

*** It is not clear to me why the dependency of \Phi in t has disappeared in Theorem 2. It is not clear either in your statement whether \Phi is optimal at all for the problem in Theorem 1.

*** the "semi-backward Euler method" is introduced without any context. The fact that it is presented as a proposition using qualitative qualifiers such as "sufficient regularity" is suspicious.

---

> ### Author Response · Authors · 2018-11-17
> **Response to AnonReviewer3**
>
> Summary of the response: AnonReviewer3 misunderstands important parts of the paper.
>
> The reviewer misinterprets the relations between distance and the gradient operator. We will explain these relations below. We first present the detailed comments to reviewer 3’s questions.
>
> 1. Q: *** "Wasserstein-2 distance on the full density set": what do you mean exactly? that d_W(\theta_0,\theta_1) \ne W(p_{\theta_0},p_{\theta_1})? Could you elaborate where this analogy breaks down?
>
> Answer: Distance on the full density set is the distance measured when we are free to move without any constraints on the tangent direction. When we have a parametrized set of densities (e.g., a neural network), the Wasserstein-2 induced metric is constrained, as we can only move along the set of densities from our parameterized set.
>
> For a visual example, imagine measuring the distance between two points in 3D by the length of a string spanned between the two points vs. measuring the distance between two points on a sphere in 3D by the length of a string on the sphere connecting the two points. The two distances are different.
>
> When and how the constrained Wasserstein metric is different from the one in the full density set has been studied in Theorem 7 and Proposition 8 of [arXiv:1803:07033]. It is mainly based on the derivation of the second fundamental formula in the Wasserstein geometry. The proof is as follows: One checks that the constrained metric tensor coincides with the one for the unconstrained set. It is to show that the second fundamental formula equals zero, for a given probability model, e.g. following the derived second fundamental formula, one can check that the Wasserstein metric constrained in the mixtures of Gaussians are not geodetically complete.
>
> There are some cases where the distance in the full set equals the distance in a constrained set (known as the totally geodesic submanifold). For Wasserstein-2 metric, it is well known that the set of Gaussian distributions has this property. For details, we refer the reader to [arXiv:0801.2250]. One can check that it simply satisfies Proposition 8 of [arXiv:1803:07033], in which the second fundamental form equals zero, which proves that Gaussian measures are totally geodesic submanifold.
>
> We are happy to add comments about this in the revised paper to clarify these relations.
>
> 2.Q: *** It is not clear to me why the dependency of \Phi in t has disappeared in Theorem 2. It is not clear either in your statement whether \Phi is optimal at all for the problem in Theorem 1.
>
> Answer: These two questions relate to the definition of metric structure. We address them in turn.
>
> a) The metric introduces a norm in the tangent space. The metric norm is defined at each point, which should not depend on time. The gradient operator is also defined at each point, and should not depend on the time either.
> For example, for any metric function of Theta defined by the action functional
>          d(theta_0, theta_1)^2 = inf_{theta(t)} {int_0^1 dot theta(s)^T G(theta(s)) dot theta(s) ds }.
> The metric at theta means
>              (a,b)_theta = a^T  G(theta) b, where a, b are tangent vectors at theta.
> For any theta in Theta, the Riemannian gradient is defined by
>                            Grad f(theta)= G(theta)^{-1} nabla_theta f(theta).
> In the context of the Wasserstein-2 metric, the metric tensor is the inverse of the weighted Laplacian operator. One way for representing it is through Phi, which is the dual variable in the tangent space. According to the above-mentioned definitions, Phi does not depend on time. We refer AnonReviewer3 to [arXiv: 1803.06360] for detailed explanations.
>
> b) Kindly note that Theorems 1 and 2 describe the constrained metric. This is a new optimal transport metric in the parameter space (of the generators). The constrained metric structure does not break the metric tensor structure. It just constrains to the direction that is feasible on the parameter space. We refer AnonReviewer3 to [S Amari. Natural Gradient Works Efficiently in Learning, 1998]. All the proofs locally in time of the unconstrained setting still works here. Theorem 1 pulls back the optimal transport metric tensor to the parameter space. For more related discussions, we refer AnonReviewer3 to [arXiv:1803:07033].

---

> ### Author Response · Authors · 2018-11-17
> **Response to AnonReviewer3 continued**
>
> 3. Q:*** the "semi-backward Euler method" is introduced without any context. The fact that it is presented as a proposition using qualitative qualifiers such as "sufficient regularity" is suspicious.
>
> Answer: The semi-backward Euler method is proposed as an effective approximation to the Wasserstein proximal operator. The motivation for this is that solving the backward method involves solving the entire path for the constrained metric. This can be hard in numerics. The time reparameterization and semi-backward Euler idea allow us to derive a simple computational method. It just introduces a simple primal-dual minimization for the computation of the gradient operator. These details are available in any book on numerical methods for differential equations, including the backward Euler and semi-backward Euler method.
>
> Sufficient regularity is for the rigorous mathematical statement. For the simplicity of presentation, we omitted the details. The current space of Phi requires knowledge of the generator, in which (nabla Phi)\in L^2(\rho(theta_k)). In later computations, we apply neural networks to solve the problem. This becomes a standard finite dimensional maximation. As long as the network model of Phi has a well defined (nabla Phi)^2, the inf-supremum problem is well posed.
>
>
> Further comments:
>
> We next comment on AnonReviewer3’s comments on the organization, description of the paper, and judgement.
>
> Organization: AnonReviewer3 thinks that the paper is ``head over heels’’. Having first motivation and then a simplified method, or first a simplified method and then motivation, is really open for debate. Yes, we could have presented a simple method first, and then the motivation. We think the paper is organized in a way that most readers should be able to navigate it with relative ease. It is also natural to first formulate a mathematical problem and general method, to then explore avenues for obtaining effective practical simplifications. The mathematical derivation provides intuition, motivation, and allows the reader to track the approximations. It also provides practical points of departure for developing further methods and alternative simplifications.
>
> Judgement: “I believe that a large pool of readers at ICLR will be extremely disappointed and frustrated to see all of this relatively arduous technical presentation produce such a simple result which, in essence, has absolutely nothing to do with the Wasserstein distance, nor with a "Wasserstein natural gradient".”
>
> One can also take the standpoint that a good paper should provide a theoretically sound and well-motivated derivation which ultimately leads to a simple and practical algorithm, which is precisely what we are doing.
>
> We are stunned about AnonReviewer3 claim that the method ``has absolutely nothing to do with the Wasserstein distance, nor with a Wasserstein natural gradient’’. This is a gross mischaracterization of the paper.
>
> It is disappointing that AnonReviewer3 claims ``The reviewer is absolutely certain that the evaluation is correct and very familiar with the relevant literature’’, when his or her comments indicate quite the contrary: “This metric defines a Riemannian metric between two parameters, by considering the resulting vector field that solves this equation.”

---

### Meta-Review · Area_Chair1 · 2018-12-18
**Reject**

**Confidence:** 4
**Recommendation:** Reject

**Metareview:**

Both R3 and R1 argue for rejection, while R2 argues for a weak accept. Given that we have to reject borderline paper, the AC concludes with "revise and resubmit".